# Delivery Outcome of Fetuses with Congenital Heart Disease—Is It Influenced by Prenatal Diagnosis?

**DOI:** 10.3390/jcm11144075

**Published:** 2022-07-14

**Authors:** Alina Weissmann-Brenner, Noam Domniz, Raanan Meyer, Tal Weissbach, Tal Elkan, Yishai Salem, Yossi Bart, Eran Kassif, Boaz Weisz

**Affiliations:** 1Institute of Obstetrical and Gynecological Imaging, The Department of Obstetrics and Gynecology, The Chaim Sheba Medical Center, Tel Hashomer, Ramat-Gan 52621, Israel; domnizn@clalit.org.il (N.D.); meyerr@gmail.com (R.M.); ferbyt@gmail.com (T.W.); elkant@gmail.com (T.E.); barty@gmail.com (Y.B.); eranka@gmail.com (E.K.); boaz.weisz@sheba.health.gov.il (B.W.); 2The Sackler School of Medicine, Tel Aviv University, Tel Aviv 6997801, Israel; yishai.salem@sheba.health.gov.il; 3Pediatric Cardiology Unit, Department of Pediatrics, The Chaim Sheba Medical Center, Tel Hashomer, Ramat-Gan 52621, Israel

**Keywords:** congenital heart disease, ultrasound, delivery, non-reassuring fetal heart rate

## Abstract

**Objective:** The objective of this study is to assess the delivery outcomes of neonates with congenital heart defects (CHD), and to explore the effect of prenatal diagnosis on these outcomes. **Methods:** A retrospective study including singleton deliveries between 2011 and 2020. All singleton neonates delivered at >24 weeks of gestation were included in this study. Fetuses with known prenatal anomalies other than CHD were excluded from this study. Pregnancy and neonatal outcomes were analyzed. A comparison was made between pregnancies with CHD and controls; and between pregnancies with prenatal diagnosis of CHD and postnatal diagnosis of CHD. **Results:** A total of 1598 neonates with CHD (688, 43.1% diagnosed prenatally) comprised the study group, compared to 85,576 singleton controls. Pregnancies with CHD had significantly increased BMI before pregnancy, suffered more from diabetes and chronic hypertension, had more inductions of labor, and had more cesarean deliveries (CD) including both elective CD and urgent CD due to non-reassuring fetal monitor (NRFHR) (OR = 1.75; 95%CI 1.45–2.14). Prenatal diagnosis of CHD is associated with a significant increased rate of induction of labor compared to postnatal diagnosis of CHD (OR = 1.59; 95% CI 1.15–2.22), but did not affect the mode of delivery including the rate of CD and CD due to non-reassuring fetal heart rate (NRFHR). Gestational age at birth and birthweight were significantly lower in pregnancies with CHD compared to controls, with no difference between prenatal to postnatal diagnosis of the anomaly. Neonates with CHD had a higher incidence of hypoxic ischemic encephalopathy and seizures compared to controls without any impact by prenatal diagnosis. **Conclusion:** Prenatal diagnosis of CHD is associated with an increased rate of induction of labor, with no increased rate of CD and CD due to NRFHR. The 5-min Apgar score is lower in pregnancies with postnatal diagnosis of CHD.

## 1. Introduction

The prenatal diagnosis of congenital heart defects (CHD) allows better planning of perinatal management in order to improve both neonatal and long-term outcomes [1]. Gestational age at delivery, mode of delivery and place of delivery are influenced by the risk of hemodynamic instability at birth [2], depending on whether the defect is shunt dependent, whether there is cardiac dysfunction, and whether there is an associated abnormality of the respiratory system [3]. The majority of children with prenatal diagnosis of CHD are stable until birth and do not require any specialized care in the prenatal period. In cases where a specialized cardiac team is needed after delivery, the location of delivery depends on the availability of a pediatric cardiac unit. Planned induction at term is considered in CHD with a minimal risk of hemodynamic instability, and in cases with likely hemodynamic instability. Cesarean delivery (CD) is considered in selected cases when coordination is needed between the obstetricians and the cardiologists, and in cases of CHD with expected hemodynamic instability. Previous studies concluded that, in most cases, labor is safe for fetuses with CHD [4,5].

The objectives of the present study were (1) to assess the delivery and short-term neonatal outcome of fetuses with cardiac anomalies, and (2) to analyze whether prenatal diagnosis (and physician awareness during delivery) of the anomaly affects the mode of delivery in these cases.

## 2. Methods

A retrospective cohort study was performed on all pregnancies with a prenatal or postnatal diagnosis of CHD who delivered in our medical center between 2011 and 2020. 

The study protocol was approved by our local Institutional Review Board at the Sheba Medical Center (No. SMC 5344–18).

The following data were extracted: Maternal characteristics including age, body mass index (BMI), substance abuse (including smoking and alcohol), pregestational diabetes mellitus, mode of conception, gravidity and parity. Data regarding pregnancy and pregnancy outcomes included gestational diabetes mellitus, hypertensive disorders of pregnancy, gestational age at delivery, mode of delivery, presence of meconium-stained amniotic fluid and fetal death. Neonatal characteristics included birthweight, neonatal gender, 5-min Apgar score, admission to neonatal intensive care unit (NICU), convulsions, blood transfusions, phototherapy due to hyperbilirubinemia, hypoglycemia and need for mechanical ventilation.

Mode of delivery was characterized as spontaneous vaginal delivery, operative vaginal delivery or cesarean delivery (CD) with a sub-analysis of CD due to non-reassuring fetal heart rate (NRFHR). Suspected fetal acidemia was characterized as either one of the following—cord pH ≤ 7.0 or base deficit ≥ 12 mmol/L [6]. Hypoxic ischemic encephalopathy (HIE), neonatal seizures and need for cooling protocol were also analyzed. Modified adverse neonatal outcome (mANO) was characterized as either one of these four entities (acidemia, HIE, neonatal seizures and cooling protocol). Since 1-min Apgar score, need for mechanical ventilation as well as admission to NICU might be biased in cases with cardiac abnormalities, they were not included in the definition of mANO. However, evaluation of 5-min Apgar score was also performed and evaluated separately [7,8].

A comparison was made to a control group that included all pregnancies without any structural anomaly and a secondary analysis was performed in CHD cases between those who were diagnosed prenatally to those without prenatal diagnosis. 

Statistical analysis was performed by SPSS statistics (IBM). Normality of the data was tested using the Shapiro–Wilk test or the Kolmogorov–Smirnov test. Data are presented as the percent and numbers or mean and standard deviation, as appropriate. Comparison between two unrelated variables was conducted with Student’s *t*-test or the Mann–Whitney *U* test, as appropriate. The Chi-Square test and Fisher’s Exact test were used for comparison between categorical variables. 

Significance was determined at *p* < 0.05 and borderline significance at *p* = 0.05–0.1. Statistical analyses were conducted using IBM SPSS v.25(IBM Corporation Inc., Armonk, NY, USA).

## 3. Results

During the study period, there were a total of 95,581 deliveries > 24^0/7^ weeks at our medical center. Cases with non-cardiac major structural anomalies or known genetic abnormalities and multi-fetal pregnancies were excluded. Overall, 1598 neonates with CHD (688, 43.1% were diagnosed prenatally) comprised the study group, compared to 85,576 singleton controls. Cardiac anomalies were characterized based on the definitions of Gindes et al. [9] divided into 4-chamber view anomalies (*n* = 1150; including, VSD, ASD, AV canal, and hypoplastic left/right heart), outflow-tract anomalies (*n* = 213; including aortic and pulmonic stenosis, tetralogy of Fallot, transposition of great arteries and truncus anomalies), arch anomalies (*n* = 179 including coarctation of aorta, interrupted aortic arch, double aortic arch and right aortic arch), vein anomalies (*n* = 7; including persistent left superior vena cava and interrupted IVC), persistent rhythm anomalies (*n* = 149), and situs anomalies (*n* = 10; situs inversus and dextrocardia). One-hundred and ten neonates (6.5% of all CHD cases) had multiple cardiac anomalies. The characteristics of pregnancies with CHD are presented in Table 1. 

Pregnancies with CHD had significantly increased BMI and increased weight before pregnancy and suffered more from diabetes, both pregestational and gestational, and from chronic hypertension. The rate of non-spontaneous conception (either ovulation induction or in vitro fertilization) was also higher in study group. 

Characteristics of mode of delivery are presented in Table 2. Pregnancies with CHD had more inductions of labor, more cesarean deliveries including both elective cesarean deliveries and urgent cesarean deliveries due to NRFHR (OR = 1.75; 95%CI 1.45–2.14). Prenatal diagnosis of CHD were associated with a significant increased rate of induction of labor compared to pregnancies with postnatal diagnosis of CHD (OR = 1.59; 95% CI 1.15–2.22), but it did not have any effect on the mode of delivery including the rate of operative vaginal deliveries, nor the rate of CD and CD due to NRFHR (*p* = 0.921). Gestational age at birth and birthweight was significantly lower in pregnancies with CHD compared to controls, but there was no difference in gestational age at delivery and in birthweight between pregnancies with prenatal diagnosis of CHD compared to postnatal diagnosis of CHD. A multiple logistic regression (75,371 deliveries—after excluding cases delivered by elective CD without trial of labor) found that the following parameters were significantly associated with CD due to NRFHR: cardiac anomaly (*p* < 0.001), primiparity (*p* < 0.001), pregestational diabetes (*p* = 0.016), prematurity (delivery < 37 weeks’ gestation) (*p* < 0.001), induction of labor (*p* = 0.001) and pregnancy conceived by artificial reproductive technology (*p* = 0.003). BMI, chronic hypertension, gestational diabetes and prenatal diagnosis of the cardiac anomaly were not associated with CD due to NRFHR.

The neonatal outcomes are presented in Table 3 and Table 4. Since cardiac anomaly is associated by itself with an increased risk of hospitalization in NICU and need for mechanical ventilation, these parameters were not included in the neonatal analysis. There was a trend for increased incidence of acidemia in neonates with CHD. Neonates with CHD had a significantly higher incidence of HIE, seizures and mANO compared to controls without any impact by prenatal diagnosis. The rate of a 5-min Apgar score < 7 was significantly higher in the CHD group compared to controls; prenatal diagnosis was associated with a slightly lower rate of Apgar (5 min) < 7. A multiple logistic regression (after excluding cases delivered by elective CD) found that the following parameters were significantly associated with mANO: cardiac anomaly (*p* = 0.003), primiparity (*p* < 0.001), gestational diabetes (*p* < 0.001), preterm labor (*p* < 0.001) and CS due to NRFHR (*p* < 0.001). Prenatal diagnosis of CHD, induction of labor, pregestational DM, chronic HTN and BMI were not associated with mANO.

Neonates with CHD who underwent elective cesarean delivery had a similar neonatal outcome to those who were intended for vaginal delivery (Table 4). 

## 4. Discussion

The main findings of our study were:Pregnancies with fetal CHD are significantly more prone to CD (31.7% vs. 24.3%) and CD due to NRFHR (7.0% vs. 4.0%) independent of prenatal diagnosis of the anomaly.Neonates with CHD are more prone to adverse neonatal outcomes such as 5-min Apgar score < 7, seizures and hypoxic ischemic encephalopathy (1.6% vs. 0.7%) independent of prenatal diagnosis of the anomaly.

Advances in prenatal imaging in fetal cardiology improved the examination of the fetal cardiovascular system leading to more accurate prenatal diagnosis of CHD [1]. Nevertheless, many patients do not perform level II anatomical scans during pregnancy, while others are not diagnosed prenatally, leading to the diagnosis of many of the cardiac anomalies only after birth [10].

In concordance to previous studies, we found that pregnancies with CHD had significantly increased pre-pregnancy BMI, were associated with assisted reproductive treatments and had significantly more hypertensive disorders, pregestational and gestational Diabetes [2]. As reported before [2,4,11,12,13,14], pregnancies with CHD were associated with an increased risk of preterm delivery (OR = 1.79; 95%CI = 1.52–2.10). Similarly, Walsh et al. [4] and Tanner et al. [11] reported a slightly higher odds ratio of 3.4 and 2.4, respectively, of preterm deliveries in pregnancies with CHD. Castellanos et al., reported 18.7% of pregnancies with critical CHD were delivered preterm. However, in our study, there was no difference in gestational age at delivery, the rate of preterm deliveries or birthweight between pregnancies with prenatal diagnosis of CHD compared to postnatal diagnosis of CHD. 

The present study, performed on a large population, demonstrated that pregnancies with CHD undergo more cesarean deliveries including both elective CD and urgent CD due to NRFHR. The prenatal awareness of the anomaly resulted in significantly more inductions of labors, but this had no influence on the mode of delivery, nor on the rate of urgent CD due to NRFHR. 

It is considered that the majority of fetuses with CHD are stable at birth and do not require any specialized care in the perinatal period, making the place of delivery and the timing of delivery determined according to obstetrical considerations. In cases with a suspected risk of neonatal cardiac compromise, delivery is planned near a specialized cardiac center to ensure the neonate receives all the specific needed cardiac interventions [1,4,12,15]. Sanapo et al., divided neonates with CHD according to the risk of hemodynamic instability at birth. They recommended planned induction to pregnancies with any risk of hemodynamic instability. Indeed, in our study, prenatal diagnosis of CHD resulted in a slightly increased rate of inductions of labor (OR = 1.28; 95%CI = 1.08–1.51) [1,3]. Similarly, Peyvandi et al., compared 86 complex CHD diagnosed prenatally to 10 diagnosed postnatally, and demonstrated that 46% had induction of labor at 39 without leading to earlier term deliveries and lower birthweight [16]. Although there were more inductions of labor in pregnancies diagnosed prenatally with CHD compared to the cases diagnosed postnatally, no difference was found in the mode of delivery. 

In this study, there was an increased risk of CD due to NRFHR (OR = 1.75; 95%CI = 1.44–2.13) in pregnancies with CHD, both when diagnosed prenatally or postnatally. Similarly, Walsh et al., in their study on 126 pregnancies with CHD, calculated a relative comparable odds ratio of 2.2 to intrapartum CD due to NRFHR [4]. Myooshi et al., reported the 18.6% rate of CD due to NRFHR in 199 singleton pregnancies with CHD who attempted vaginal delivery. They found that fetal heart failure, low birth weight and primiparity status were independent predictors of urgent CD due to NRFHR in CHD infants [17]. A multiple logistic regression of our results found as well that CHD, primiparity and prematurity were associated with CD due to NRFHR. Our study also showed that pregestational diabetes and non-spontaneous conception were also independently associated with CD due to NRFHR.

Neonates with CHD had significantly more HIE, seizures and mANO (including a trend for more acidemia). This too may be partially explained by the congenital cardiac anomaly that may influence the neonatal outcomes. Indeed, Boos et al., reported 3.4% of perinatal asphyxia in 504 patients with ductal dependent CHD [7]. Prenatal diagnosis of the anomaly did not influence the rate of mANO, nor the rate of HIE or neonatal seizures. The rate of elective CD (without trial of labor) was higher in the CHD group (16.6% vs. 13.4%, *p* < 0.001). However, delivery by elective CD in pregnancies with CHD did not reduce significantly the rate of HIE, seizures or mANO. 

Significantly more pregnancies with CHD had a 5-min Apgar score < 7, but there was no significant difference in the cord PH. This can be explained in part by the cyanotic appearance of neonates with cyanotic CHD, or the lower pulse in CHD with bradyarythmias. Similar to our study, Adams et al., in their study on 134 pregnancies with CHD, demonstrated that there was no difference in the umbilical artery PH in the CHD group vs. the control group. They concluded that the circulatory alterations that occur as a result of CHD are well tolerated in utero [18,19]. Levey et al., in their study on 439 neonates with CHD, found no difference in Apgar scores between prenatal and postnatal diagnosis [14]. Contrary to that, our study demonstrated that significantly more pregnancies with postnatal diagnosis of CHD had a 5-min Apgar lower than 7 compared to the prenatally diagnosed. This can be explained with the careful follow up of these pregnancies including the induction of delivery and continuous monitoring during labor of these deliveries.

This is the largest-scale population study evaluating delivery outcome and neonatal outcome in pregnancies with CHD compared to controls and a further analysis of prenatal diagnosis on these outcome variables. In summary, pregnancies with CHD had more CD and CD due to NRFHR. They were also associated to increased rate adverse neonatal outcome. However, these outcome parameters were not influenced by prenatal diagnosis and knowledge of the anomaly.

We acknowledge several limitations in our study. The retrospective nature of our study carries limitations inherent to retrospective investigations. Furthermore, we analyzed all cases of CHD without categorizing to the subtypes according to complexity, hemodynamic instability or association to cyanosis. Moreover, the distribution of different types of cardiac anomaly is not similar between the pre- and postnatal diagnosis groups. Further analysis is needed to compare the pregnancy outcomes according to the different categories of CHD.


**What are the novel findings of this work?**


This is the first study to report a higher incidence of cesarean deliveries (CD) due to non-reassuring fetal heart rate (NRFHR) in cases with fetal congenital heart defect (CHD) unrelated to prenatal diagnosis of the anomaly (and therefore biased management of birth). 


**What are the clinical implications of this work?**


These findings highlight the association of CHD with an increased risk of cesarean deliveries and CD due to NRFHR as well as an increased risk of adverse neonatal outcome. 

## Figures and Tables

**Table 1 jcm-11-04075-t001:** Characteristics of 87,174 deliveries > 24^0/7^ weeks. Comparison between controls and fetuses with congenital cardiac anomalies.

	Cardiac Anomaly (*n* = 1598)	Control (*n* = 85,576)	
Gravidity	2 (1–19)	2 (1–20)	**0.034**
Parity	1 (0–14)	1 (0–14)	**0.001**
Primipara	536 (33.5%)	31,018 (36.2%)	**0.026**
Maternal age	32.3 ± 5.5	32.1 ± 5.1	0.838
Weight (before pregnancy)	63.75 ± 13.72	62.63 ± 12.90	**0.003**
BMI (before pregnancy)	23.9 ± 6.6	23.4 ± 6.4	**0.003**
BMI (before delivery)	26.6 ± 8.3	26.7 ± 7.6	0.546
Spontaneous pregnancy	1443 (90.3%)	75906 (88.7%)	**0.046**
Diabetes—pregestational	41 (2.5%)	1519 (1.7%)	**0.022**
Diabetes—gestational	192 (12.0%)	8317 (9.7%)	**0.003**
Hypertensive disease of pregnancy (PIH/PET)	68 (4.2%)	3670 (4.2%)	1.0
Chronic HTN	26 (1.6%)	901 (1.0%)	**0.035**

**Table 2 jcm-11-04075-t002:** Mode of delivery of 87,174 singleton pregnancies born after 24^0/7^ weeks of gestation *-(%) of all spontaneous onset of labor.

	Cardiac Anomaly (1598)	Control (85,576)	*p*	Prenatal Diagnosis of Cardiac Anomaly (688)	Postnatal Diagnosis of Cardiac Anomaly (910)	*p*
GA at delivery mean (SD)	38 + 6/7 (2 + 1/7)	39 + 2/7 (1 + 6/7)	**<0.001**	38 + 6/7 (2 + 3/7)	38 + 5/7 (1 + 6/7)	0.194
Birthweight	3111 ± 601	3219 ± 493	**<0.001**	3111 ± 582	3111 ± 615	0.996
Preterm delivery (<37 w)	10.9%	6.4%	**<0.001**	11.4%	10.2%	0.238
Spontaneous onset of labor	694 (43.4%)	45331 (52.9%)	**<0.001**	283 (41.1%)	411 (45.2%)	0.114
Induction of labor	157 (9.8%)	6713 (7.8%)	**0.002**	84 (12.2%)	73 (8.0%)	**0.006**
Elective CD	266 (16.6%)	11537 (13.4%)	**<0.001**	119 (17.3%)	147 (15.8%)	0.732
ECD for maternal request	18 (1.1%)	1069 (1.2%)	0.819	10 (1.5%)	8 (0.9%)	0.341
Mode of delivery: Spontaneous Operative vaginal delivery CD	1010 (63.2%) 81 (5.0%) 507 (31.7%)	60102 (69.5%) 5360 (6.2%) 21,007 (24.3%)	**<0.001** **0.05** **<0.001**	433 (62.9%) 38 (5.2%) 219 (31.8%)	577 (63.6%) 43 (4.8%) 288 (31.6%)	0.987 0.476 0.980
CD for NRFHR	111 (7.0%)	3487 (4.0%)	**<0.001**	47 (6.8%)	64 (7.0%)	0.921

**Table 3 jcm-11-04075-t003:** Neonatal outcome of newborns with cardiac anomaly (prenatal vs. postnatal diagnosis). NRFHR: non-reassuring fetal heart rate.

	Cardiac Anomaly (1598)	Control (85,576)	*p*	Prenatal Diagnosis of Cardiac Anomaly (688)	Postnatal Diagnosis of Cardiac Anomaly (910)	*p*
Cord pH < 7.0Or BE < −12	14 (0.9%)	490 (0.6%)	0.08	6 (0.9%)	8 (0.9%)	0.605
Cooling protocol	3 (0.2%)	62 (0.1%)	0.117	1 (0.1%)	2 (0.2%)	0.604
HIE	6 (0.4%)	77 (0.1%)	**0.004**	1 (0.1%)	5 (0.5%)	0.188
Seizures	9 (0.6%)	86 (0.1%)	**<0.001**	2 (0.3%)	7 (0.8%)	0.315
mANO (Any of the above)	25 (1.6%)	602 (0.7%)	**<0.001**	7 (1.0%)	18 (2.0%)	0.155
5 min Apgar < 7	22 (1.4%)	261 (0.3%)	**<0.001**	3 (0.4%)	19 (2.1%)	**0.003**
Phototherapy	159 (9.9%)	5666 (6.6%)	**<0.001**	57 (8.3%)	102 (11.2%)	0.063
Blood transfusion	60 (3.8%)	305 (0.4%)	**<0.001**	25 (3.6%)	35 (3.8%)	0.895
Hypoglycemia	157 (9.8%)	4515 (5.3%)	**<0.001**	47 (6.8%)	64 (7.0%)	0.921

**Table 4 jcm-11-04075-t004:** Neonatal outcome for singleton neonates with cardiac anomaly (all vs. Elective CD).

	Cardiac Anomaly Singleton			*p*
	All (1598) (Gr.1)	Non-ECD (1335)	ECD (263)	ECD vs. Non-ECS (all)
Cord pH < 7.0 Or BE < −12	14 (0.9%)	14 (1.0%)	0 (0%)	0.145
Cooling protocol	3 (0.2%)	2 (0.2%)	1 (0.1%)	1.0
HIE	6 (0.4%)	5 (0.2%)	1 (0.0%)	1.0
Seizures	9 (0.6%)	8 (0.6%)	1 (0.4%)	1.0
mANO (Any of the above)	25 (1.6%)	22 (1.6%)	3 (1.1%)	0.786
5 min Apgar < 7	22 (1.4%)	19 (1.4%)	3 (1.4%)	1.0

## Data Availability

The data that support the findings of this study are available on request from the corresponding author. The data are not publicly available due to privacy or ethical restrictions.

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
