# Peer review of "Delivery Outcome of Fetuses with Congenital Heart Disease—Is It Influenced by Prenatal Diagnosis?"

_jcm, 2022, doi:10.3390/jcm11144075_

Round 1

Reviewer 1 Report

This is an article trying to evaluate if delivery outcomes are influenced by prenatal diagnosis of congenital heart disease. The authors have performed a retrospective study from a large database of over 85000 deliveries. A few suggestions to incorporate in the article are as follows:

Abstract:

The authors mention that prenatal diagnosis of CHD resulted in increased rate of induction of labor. This may need to be changed to "associated with" since the authors do not report the reasons for IoL

The authors mention that apgar score <7 was higher in CHD compared to controls but this was ameliorated by prenatal diagnosis. Again this is an association as it is difficult to prove causation from a retrospective study.

Typos and grammatical errors need to corrected.

Introduction:

Objectives: Please specify what neonatal outcomes are being evaluated. From the article it appears that the authors are only looking at outcomes related to delivery room management of neonates and not short or long term outcomes. 

Methods: 

Its critical for this study to mention the types of CHD that were included and to evaluate the data stratifying based on severity of the disease. 

The authors evaluated several neonatal characteristics such as blood transfusions, hyperbilirubinemia, hypoglycemia but provide no data in the results section or make a mention of this anywhere in the article other than in methods. 

Need to elaborate on the statistical analysis.

Results: 

Table 1: What p-value was considered as significant? If the conventional <0.05 was considered then pregestational diabetes does not appear to be significant.

Table 2: Please provide SD for GA for Gr. 2 and Gr. 3

Table 2: Can the table be split into 2 tables or better organized to make it less cluttered? It could be something like Cases vs controls and prenatally diagnosed vs postnatally diagnosed.

What are the reasons for induction of labor? The authors state that prenatal diagnosis of CHD resulted in increased induction of labor. Did you look at specific reasons for IoL? or was this adjusted statistically? Some clarity needs to be provided. As it stands, it just appears to be an association.

Authors mention they performed regression analysis after excluding cases with elective CD without trial of labor.  Who were included in this analysis and what were the total number of deliveries. Need clarity on this data.

When commenting about 5minute apgar scores, the authors state again that the score <7 was ameliorated by prenatal diagnosis of CHD. This should be changed to association.

Table 3: Why is apgar <7 at 5minutes chosen as an indicator of adverse neonatal outcome? It may be better and more reliable to use 5minute apgar as a continuous variable and evaluate if there was a statistical difference in the median(range) 5minute apgar score. 

Discussion:

In the main findings

Point 2: it is unclear what is being communicated. Please rephrase.

In paragraph 7: 5minute apgar score is stated to be significantly lower in pregnancies with CHD. But data shown is of Apgar score <7 at 5 minutes of life. These are not the same. Please maintain consistency. 

One of the disadvantages of the study that the authors acknowledge is that the cases were not analyzed based on category of CHD.  I think this is an important aspect that this study is lacking. Providing this data will probably improve this manuscript significantly. If possible please provide this data. 

Author Response

Reviewer no. 1:

This is an article trying to evaluate if delivery outcomes are influenced by prenatal diagnosis of congenital heart disease. The authors have performed a retrospective study from a large database of over 85000 deliveries. A few suggestions to incorporate in the article are as follows:

Abstract:

Comment #1: The authors mention that prenatal diagnosis of CHD resulted in increased rate of induction of labor. This may need to be changed to "associated with" since the authors do not report the reasons for IoL.

Reply to comment #1: Thank you, we changed the sentence as requested.

Comment #2: The authors mention that Apgar score <7 was higher in CHD compared to controls but this was ameliorated by prenatal diagnosis. Again this is an association as it is difficult to prove causation from a retrospective study.

Reply to comment #2: Indeed, and this is why we have now decided to erase the sentence from the Abstract section.

Comment #3: Typos and grammatical errors need to corrected.

Reply to comment #3: Grammatical corrections were made in the entire manuscript as requested. 

Comment #4: Introduction: Objectives: Please specify what neonatal outcomes are being evaluated. From the article it appears that the authors are only looking at outcomes related to delivery room management of neonates and not short or long term outcomes. 

Reply to comment #4: In our results section (as well as Tables No 3 and 4) we included also neonatal outcome such as HIE, neonatal seizures and need for cooling- therefore we think that the preferred term is " to assess the delivery and short-term neonatal outcome of fetuses with cardiac anomalies". However, if needed "delivery outcome" is also suitable.  

Methods: 

Comment #5: Its critical for this study to mention the types of CHD that were included and to evaluate the data stratifying based on severity of the disease. 

Reply to comment #5: Thank you for your comment. Details about the types of CHD were included in the results section: "Cardiac anomalies were characterized based on the definitions of Gindes et al (Gindes L, Hegesh J, Weisz B, Gilboa Y, Achiron R. Three and four dimensional ultrasound: a novel method for evaluating fetal cardiac anomalies. Prenatal Diagnosis. 2009;(January):645–53) divided into 4-chamber view anomalies (n=1150, prenatal diagnosis=43.7%; including, VSD, ASD, AV-canal, Hypoplastic left/right heart), Outflow-tract anomalies (n= 213, prenatal diagnosis=27.2%; including aortic and pulmonic stenosis, tetralogy of Fallot, transposition of great arteries and truncus anomalies), Arches anomalies (n= 179, prenatal diagnosis=84.9%; including coarctation of aorta, interrupted aortic arch, double aortic arch and right aortic arch), Vein anomalies (n=7, prenatal diagnosis 4/7; including persistent left superior vena cava and interrupted IVC) and persistent rhythm anomalies (n=149), Situs anomalies (n=10, prenatal diagnosis =70%; situs inversus and dextrocardia). One-hundred and ten neonates (6.5% of all CHD cases) had multiple cardiac anomalies." Unfortunately, the severity of each anomaly (specifically for the anatomical abnormality found and its variations) is not detailed in our database- for example- When TOF was diagnosed- our database does not include the diameter of the pulmonary arteries- etc.

Comment #6: The authors evaluated several neonatal characteristics such as blood transfusions, hyperbilirubinemia, hypoglycemia but provide no data in the results section or make a mention of this anywhere in the article other than in methods. 

Reply to comment #6: As suggested we have incorporated neonatal data including the need for phototherapy due to jaundice, neonatal blood transfusions and hypoglycemia to table 3 showing increased incidence of these events in neonates with CHD , although not related to prenatal diagnosis.

Comment #7: Need to elaborate on the statistical analysis.

Reply to comment #7: As suggested, statistical analysis was detailed in Method section.

Results: 

Comment #8: Table 1: What p-value was considered as significant? If the conventional <0.05 was considered then pregestational diabetes does not appear to be significant.

Reply to comment #8: We thank the reviewer for this helpful comment- since this has driven us to check stats again and it seems we have had a mistake and the 2-sided p value is 0.02. This was altered in the text.

Comment #9: Table 2: Please provide SD for GA for Gr. 2 and Gr. 3

Reply to comment #9: As suggested, SD was added to gestational age in Table 1.

Comment #10: Table 2: Can the table be split into 2 tables or better organized to make it less cluttered? It could be something like Cases vs controls and prenatally diagnosed vs postnatally diagnosed.

Reply to comment #10: As suggested, we have organized table no 2 to make it tidier. Thanks to the reviewer's comment we also altered Table No 3 so they will look similar.

Comment #11: What are the reasons for induction of labor? The authors state that prenatal diagnosis of CHD resulted in increased induction of labor. Did you look at specific reasons for IoL? or was this adjusted statistically? Some clarity needs to be provided. As it stands, it just appears to be an association.

Reply to comment #11: Unfortunately, the reasons for induction are not present in the computerized patient data- therefore we had to change the wording of our findings accordingly to "prenatal diagnosis of CHD is associated (and not result) with significant increased rate of inductions of labor. " (Abstract) and "Prenatal diagnosis of CHD were associated with a significant increased rate of induction of labor compared to pregnancies with postnatal diagnosis of CHD" (Results).

Comment #12: Authors mention they performed regression analysis after excluding cases with elective CD without trial of labor.  Who were included in this analysis and what were the total number of deliveries. Need clarity on this data.

Reply to comment #12: As suggested, we have clarified in the results that the regression analysis was performed on 75371 patients who were not scheduled for elective CS.

Comment #13: When commenting about 5minute Apgar scores, the authors state again that the score <7 was ameliorated by prenatal diagnosis of CHD. This should be changed to association.

Reply to comment #13: Changed we made as requested.

Comment #14: Table 3: Why is Apgar <7 at 5minutes chosen as an indicator of adverse neonatal outcome? It may be better and more reliable to use 5minute Apgar as a continuous variable and evaluate if there was a statistical difference in the median(range) 5minute Apgar score. 

Reply to comment #14: We preferred using the value 5-minutes Apgar scores<7 as an indicator of adverse neonatal outcome similar to previous studies. (Daniel Shai et al. Predictive value of new onset versus primary meconium-stained amniotic fluid . Birth. 2022 May 13.  doi: 10.1111/birt.12648. Online ahead of print. Etc.) Furthermore, since cardiac anomaly per-se might be associated with slightly decreased apgar score even in non-asphyxia cases we did not find it appropriate to describe the mean 5-min Apgar score. However, this analysis was performed and the results of such an analysis show that cardiac anomaly is associated with significantly statistical (but not clinically relevant) lower Apgar score 9.77±0.85 vs. 9.89±0.74 (p<0.01) and this was also valid to patients who had elective CS: 9.78±0.46 vs. 9.93±0.46 (p<0.01). Within the cardiac anomaly group- prenatal diagnosis was not associated with significantly different Apgar score 9.75±0.95 vs. 9.79±0.62 (p=0.282).   To conclude this comment- evaluation 5-min Apgar score as a continuous variable showed similar results to evaluation of dichotomous variable in all aspects except that in the evaluation of the continuous variable there was no difference in cases with or without prenatal diagnosis. We will be happy to add this to the results section if the reviewer consider this as an important finding. 

Discussion:

Comment #15: In the main findings: Point 2: it is unclear what is being communicated. Please rephrase.

Reply to comment #15: Changes were made as requested.

Comment #16: In paragraph 7: 5minute Apgar score is stated to be significantly lower in pregnancies with CHD. But data shown is of Apgar score <7 at 5 minutes of life. These are not the same. Please maintain consistency. 

Reply to comment #16: We thank the reviewer for this comment which makes our manuscript more statistically valid (as comment #14) - Change were made as requested.

Comment #17: One of the disadvantages of the study that the authors acknowledge is that the cases were not analyzed based on category of CHD.  I think this is an important aspect that this study is lacking. Providing this data will probably improve this manuscript significantly. If possible please provide this data. 

Reply to comment #17: We thank the reviewer for this comment. We added details regarding the CHD classification in the text. Another study on the pregnancy outcomes of specific types of CHD and their severity is conducted in our medical center.

Reviewer 2 Report

This paper is well written and tackles an important issue regarding the significance of prenatal diagnosis for fetal CHD in the context of delivery pattern and post-natal stabilization. I have a few comments and questions.

  1. It is difficult to verify the benefits from prenatal dianosis of CHD in terms of postnatal outcomes. One of reasons is the difference in the severity of CHD between pre-natal and post-natal group. In this regard, the authors should elaborate on the details of CHD in both groups.
  2. In a recent study regarding the postnatal shock in babies with and without prenatal diagnosis, it was suggested that postnatal shock is more prevalent among the babies with prenatal diagnosis but persistent shock until the surgery was rare compared to the patients with postnatal diagnosis. This finding is inconsistent with the observation from this study that 5 minute APGAR score is higer among the prenatally diagnosed babies. To explain this discrepancy properly, the authors should elaborate on the severity of CHD, back to the question 1. There are two currently used CHD severity systems the authors can employ (i.e. Comprehensive Aristotle score, RACHS-1).
  3. Becuase preterm delivery is unadvisable for babies with CHD, vaginal progesteron theryapy is recommended if prenatal diagnosis of CHD is made for pregnancy at a higher risk of preterm labor. Given the gestational age of the babies with CHD, some mothers might have taken this treatment. How many pregnant mothers of the babies with prenatal diagnosis of CHD took this therapy to deliverately maintain the pregnancy up to term?

Author Response

Reviewer no. 2:

This paper is well written and tackles an important issue regarding the significance of prenatal diagnosis for fetal CHD in the context of delivery pattern and post-natal stabilization. I have a few comments and questions.

Comment #1: It is difficult to verify the benefits from prenatal diagnosis of CHD in terms of postnatal outcomes. One of reasons is the difference in the severity of CHD between pre-natal and post-natal group. In this regard, the authors should elaborate on the details of CHD in both groups.

Reply to comment #1: Details about the prenatally diagnosed CHD and the postnatally diagnosed CHD were added in the text. We agree with the reviewer that there might be difference in the severity of CHD between pre-natal and post-natal group. We could not assess the severity of the cases based on our data but we have seen that there is a difference in the distribution of anomalies between pre- and post-natal diagnosis.

*

*

The rate of outflow-tract abnormalities was significantly smaller while the rate of arch- abnormalities was significantly higher (*-p<0.05) in the prenatal diagnosis group. Obviously this might have influenced the neonatal outcome. We are planning to evaluate each cardiac anomaly solely, but this is the scope of another study. We have added this limitation to our discussion.

It is important to re-clarify that this paper does not evaluate the rate of prenatal diagnosis of cases (since many did not have an anomaly scan or fetal-echo due to religious issues) but only to determine the effect of prenatal diagnosis on neonatal outcome. 

Comment #2: In a recent study regarding the postnatal shock in babies with and without prenatal diagnosis, it was suggested that postnatal shock is more prevalent among the babies with prenatal diagnosis but persistent shock until the surgery was rare compared to the patients with postnatal diagnosis. This finding is inconsistent with the observation from this study that 5 minute APGAR score is higher among the prenatally diagnosed babies. To explain this discrepancy properly, the authors should elaborate on the severity of CHD, back to the question 1. There are two currently used CHD severity systems the authors can employ (i.e. Comprehensive Aristotle score, RACHS-1).

Reply to comment #2: We thank the reviewer for his/her comment. We added details about the CHD diagnosed prenatally and postnatally as requested. In the Discussion section we discuss the pregnancy outcomes according to the severity of the CHD: “It is considered that the majority of fetuses with CHD are stable at birth and do not require any specialized care in the perinatal period, making the place of delivery and the timing of delivery determined according to obstetrical considerations. In cases with suspected risk of neonatal cardiac compromise, delivery is planned near a specialized cardiac center to ensure the neonate receives all the specific needed cardiac interventions 1,4,11,14.

Sanapo et al (Clin Perinatol 43 (2016) 55–71 ) divided neonates with CHD according to the risk of hemodynamic instability at birth”.  In this study we did not address the outcome of cardiac surgery. We are trying to retrieve sufficient data for two projects- assessing the outcome according to Sanapo's criteria (for this we are still missing relevant details such as restriction of FO, pulmonary arteries in TOF etc)  as  well another study that is (by another student) relating this to scores of risk of CHD surgery .

Comment #3: Becuase preterm delivery is unadvisable for babies with CHD, vaginal progesteron therapy is recommended if prenatal diagnosis of CHD is made for pregnancy at a higher risk of preterm labor. Given the gestational age of the babies with CHD, some mothers might have taken this treatment. How many pregnant mothers of the babies with prenatal diagnosis of CHD took this therapy to deliberately maintain the pregnancy up to term?

Reply to comment #3: Progesterone is given to patients at higher risk for preterm deliveries, such as patients with previous preterm births. We do not usually administer progesterone for the indication of fetal CHD. We agree that it will be interesting to conduct a prospective study and examine this issue.

Reviewer 3 Report

This study investigated the perinatal prognosis of congenital heart disease. The data sample size is sufficient, the study methods are unbiased, and the evaluation of the results is reasonable.

Author Response

Reviewer no. 3:

This study investigated the perinatal prognosis of congenital heart disease. The data sample size is sufficient, the study methods are unbiased, and the evaluation of the results is reasonable.

Reply to comment: We thank the reviewer for his/her favorable assessment of the manuscript.

Round 2

Reviewer 1 Report

Thank you for the revisions.

Please proof read the manuscript and correct the errors. For eg., authors have written hemolytic ischemic encephalopathy as the expansion of HIE. 

Author Response

We thank the reviewer and have corrected the errors. 

Reviewer 2 Report

I would like to thank the authors for their efforts to revise the manuscript.  

Author Response

We would like to thank the reviewer and we have made final spell check.